# Enhanced Performance of WO_3_/SnO_2_ Nanocomposite Electrodes with Redox-Active Electrolytes for Supercapacitors

**DOI:** 10.3390/ijms24076045

**Published:** 2023-03-23

**Authors:** Tamiru Deressa Morka, Masaki Ujihara

**Affiliations:** Graduate Institute of Applied Science and Technology, National Taiwan University of Science and Technology, 43 Keelung Road, Taipei 10607, Taiwan

**Keywords:** supercapacitor, nanocomposite, energy storage, redox-additive, tungsten oxide, tin oxide

## Abstract

For effective supercapacitors, we developed a process involving chemical bath deposition, followed by electrochemical deposition and calcination, to produce WO_3_/SnO_2_ nanocomposite electrodes. In aqueous solutions, the hexagonal WO_3_ microspheres were first chemically deposited on a carbon cloth, and then tin oxides were uniformly electrodeposited. The synthesized WO_3_/SnO_2_ nanocomposite was characterized by XRD, XPS, SEM, and EDX techniques. Electrochemical properties of the WO_3_/SnO_2_ nanocomposite were analyzed by cyclic voltammetry, galvanostatic charge-discharge tests, and electrochemical impedance spectroscopy in an aqueous solution of Na_2_SO_4_ with/without the redox-active electrolyte K_3_Fe(CN)_6_. K_3_Fe(CN)_6_ exhibited a synergetic effect on the electrochemical performance of the WO_3_/SnO_2_ nanocomposite electrode, with a specific capacitance of 640 F/g at a scan rate of 5 mV/s, while that without K_3_Fe(CN)_6_ was 530 F/g. The WO_3_/SnO_2_ nanocomposite catalyzed the redox reactions of [Fe(CN)_6_]^3^/[Fe(CN)_6_]^4−^ ions, and the [Fe(CN)_6_]^3−^/[Fe(CN)_6_]^4−^ ions also promoted redox reactions of the WO_3_/SnO_2_ nanocomposite. A symmetrical configuration of the nanocomposite electrodes provided good cycling stability (coulombic efficiency of 99.6% over 2000 cycles) and satisfied both energy density (60 Whkg^−1^) and power density (540 Wkg^−1^) requirements. Thus, the WO_3_/SnO_2_ nanocomposite prepared by this simple process is a promising component for a hybrid pseudocapacitor system with a redox-flow battery mechanism.

## 1. Introduction

Clean and renewable energy is the basic criterion for sustainable development of the global economy and human lives throughout the world [1]. The massive increase in fossil fuel usage has brought two significant problems: the first is rapidly reduced fossil fuel reserves, and the second is environmental degradation, such as air/water pollution and climate change. Considering these issues, the need for renewable energy and new technology for energy storage are currently the top concerns around the world. In response to rising ecological concern and modern civilization, new environmentally friendly and cost-effective energy storage devices with excellent performance are required for customer products, such as hybrid automobiles, electronic vehicles, mobile phones, and laptop computers [2,3]. Supercapacitors, batteries, and fuel cells are the most common electrochemical energy storage devices used for this purpose. Among them, supercapacitors are emphasized because of their excellent power densities, fast charge/discharge, long-term cycling life, cost-effectiveness, good reversibility, and wide working temperature range compared to those of batteries; however, in terms of commercial application, supercapacitors are behind batteries [4,5,6]. The fundamental issue of supercapacitors is their lower energy density, especially when compared to batteries.

To solve this problem of supercapacitors, three different energy storage mechanisms have been developed: electric double-layer capacitors (EDLCs), pseudocapacitor, and battery-type capacitors [7]. Their energy storage mechanisms are explained as follows: the electric double-layer capacitor stores energy at the junction of the electrode and electrolyte by ion adsorption/desorption, the pseudocapacitor electrode stores energy on the surface of the electrode via Faradaic processes (using a fast and reversible redox process near the surface), and the battery-type capacitor has a specific potential window for the Faradaic processes used to enhance the capacitance [8]. During these electrochemical reactions, the amount of charge stored in the electrode is proportional to the total available surface of the active materials in the electrode. Therefore, porous materials with large specific surface areas are preferable for supercapacitors [1].

As active materials for pseudocapacitors, transition metal oxides (TMOs) and conducting polymers are widely used [9]. Transition metal oxides are known for high specific capacitance without sacrificing power density [10]. Among TMOs, RuO_2_, Co_3_O_4_, ZnO, MoO_3_, Fe_2_O_3_ MnO_2_, NiO, SnO_2_, WO_3_, and others have been investigated as active materials due to their appealing characteristics, such as high theoretical capacitance and low environmental impact [2,8,11,12,13]. Furthermore, composite TMOs have been designed for a pseudocapacitor with expectations of synergistic effects [6,7]. The TMOs in the composite electrode can produce unique structures with large surface areas, and the mixed valence metals can provide an extra valance state for the electrochemical process, which results in improved performance of the supercapacitor [3,5].

Recently, tungsten trioxide (WO_3_) has been considered a promising candidate for pseudocapacitors due to its numerous intrinsic features, such as high theoretical capacitance, high corrosion resistance, good chemical stability, low cost, and environmental friendliness [14], and a variety of composites with other TMOs, such as MnO_2_-WO_3_ [15], Ni-WO_3_ [16], Co-WO_3_ [17], TiO_2_/WO_3_ [18], WO_3_/Se [2], and V_2_O_5_/WO_3_ [19], have been examined. Moreover, SnO_2_ has attracted much attention as an electrode for energy storage or conversion due to its low cost, nontoxicity, high electrochemical activity, and chemical stability [20]. For successful insertion of SnO_2_ into WO_3_, excellent electrochemical supercapacitor performance is required [21,22]. Hence, motivated by the above considerations, we investigated the WO_3_/SnO_2_ nanocomposite as an active electrode material for supercapacitors.

Another approach to enhancing the capacity of the supercapacitor is to use a redox-active electrolyte. The effects of additive electrolytes, such as K_3_Fe(CN)_6_ [23,24], KI [25], and p-aminophenol [26], have been reported for aqueous systems. The basic role of these additives is to improve charge transport in the electrolyte solution, adjust the chemical state of the working electrode surface, and enhance Faradaic reactions. Synergistic effects in electrochemical processes, both in the active materials and electrolytes, can also be expected to enhance the efficiencies of redox reactions and improve the performance of supercapacitors [23]. The electrolyte additive potassium ferricyanide, K_3_Fe(CN)_6_, enables the electrode of the supercapacitor to achieve fast charging, slow discharge, and good cycling stability [27]. In this work, we investigated the preparation of a WO_3_/SnO_2_ nanocomposite through chemical bath deposition (CBD) and electrodeposition on a carbon cloth substrate, and analyzed its supercapacitor performance by using K_3_Fe(CN)_6_ added into a 1 M Na_2_SO_4_ aqueous electrolyte [28]. To the best of our knowledge, this is the first trial to synthesize a WO_3_/SnO_2_ nanocomposite and to determine its electrochemical supercapacitance performance with the redox-active electrolyte K_3_Fe (CN)_6_ with Na_2_SO_4_, and the system demonstrated a specific capacitance of 640 F/g. A symmetric supercapacitor using a WO_3_/SnO_2_ nanocomposite and an additive electrolyte was further assembled to demonstrate the potential of this system for efficient energy storage.

The reaction mechanisms for formation of WO_3_ were as follows (Equations (1) and (2)) [1,29]:(1)$${\mathrm{Na}}_{2}{\mathrm{WO}}_{4}+2\mathrm{HCl}+{\mathrm{nH}}_{2}O\;\to \;H_{2}{\mathrm{WO}}_{4}\cdot {\mathrm{nH}}_{2}O+2{\mathrm{HO}}_{4}\cdot {\mathrm{nH}}_{2}O$$
(2)$$H_{2}{\mathrm{WO}}_{4}\cdot {\mathrm{nH}}_{2}O\;\to \;{\mathrm{WO}}_{3}+(n+1)H_{2}O$$

Crystals of tungsten oxide (WO_3_) are polymorphic and often adopt a hexagonal system [1,15]. Various synthetic aspects, such as the reaction temperature, process duration, precursor type, and chemical agent (e.g., (NH_4_)_2_SO_4_), can influence the structure and morphology. In comparison to other structures, this structure provides large tunnels for ion penetration, which improves the electrochemical performance [13]. Thus, WO_3_ is usually synthesized as a hexagonal system using reagents containing sulfate ions; in particular, ammonium sulfate is preferred. The NH_4_^+^ and SO_4_^2−^ ions act as stabilizing and capping agents, respectively, during the process. The radius of the NH_4_ ion is larger than those of alkali metal ions, and these differences in size influence the penetration of numerous ions, which implies that many SO_4_^2−^ ions can readily be adsorbed on the surface parallel to the WO_3_ axis [13,30]. The morphology and crystal structure of WO_3_ are also affected by the annealing temperature. The orthorhombic phase changes into an anhydrous hexagonal phase at 400 °C and turns into stable monoclinic WO_3_ with aggregation above 500 °C [31]. Therefore, annealing in this work was done at 400 °C for 2 h.

## 2. Results

### 2.1. Characterization of WO_3_, SnO_2_, and WO_3_/SnO_2_ Nanocomposite Thin Films

The structural characteristics and crystal phases of the WO_3_, SnO_2_, and WO_3_/SnO_2_ nanocomposites were analyzed by XRD (Figure 1). The XRD peaks for WO_3_ were assigned to the hexagonal system, except for the one marked with an asterisk; this came from the carbon cloth substrate (Figure 1a). The diffraction peaks at 2θ = 14.25°, 23.28°, 28.2°, 34.8°, 37.28°, 44.30°, 50.28°, 52.40°, and 56.03° were assigned to the (100), (110), (200), (112), (202), (212), (004), (220), and (204) planes, respectively (JCPDS 85-2459). The sharp and intense peaks suggested the highly crystalline nature of the WO_3_.

The XRD pattern for the synthesized SnO_2_ showed sharp diffraction peaks at 30.7°, 32.1°, 43.9°, 45.1°, and 55.5°, which corresponded to the (221), (101), (111), (210), and (220) planes of the tetragonal phase (JCPDS 41-1445), respectively, except for the one marked with an asterisk, which came from the carbon cloth substrate (Figure 1b) [32]. Figure 1c shows the diffraction peaks for the WO_3_/SnO_2_ nanocomposite at 2θ = 14.25°, 23.28°, 28.20°, 30.71°, 32.10°, 34.80°, 37.28°, 43.20°, 44.30°, 45.10°, 50.28°, 52.40°, 55.20°, and 56.03°, which were assigned to the (100), (110), (200), (112), (202), (212), (004), (220), and (204) planes of WO_3_ and (221), (101), (111), (210), and (220) planes of SnO_2_, respectively. The diffraction peaks, denoted with diamond symbols, were indexed to the tetragonal structure of SnO_2_ [33]; similarly, the other peaks were associated with the hexagonal crystal phase of WO_3_ [1]. Therefore, this diffraction pattern was composed of characteristic peaks for both WO_3_ and SnO_2_, as observed in (Figure 1a,b), which confirmed successful preparation of the WO_3_/SnO_2_ nanocomposite. No new significant peak was observed, and thus, the WO_3_ and SnO_2_ components were separately crystallized in the nanocomposite. The presence of sharp diffraction peaks suggested the highly crystalline nature of the materials deposited on the carbon cloth substrate. The average sizes of the WO_3_ and SnO_2_ crystallites in the WO_3_/SnO_2_ nanocomposite were determined by the Scherrer equation, as shown below in (Equation (3)) [6,34,35,36]:D = K/βcosθ(3)
where D is the crystallite size (nm), β is the full width half maximum (FWHM, radians) of the peak corresponding to the plane, K = 0.9 (Sherrer constant), λ = 0.15406 nm (wavelength of the X-ray source), and θ is obtained from the 2θ value corresponding to the diffraction peak [35]. These findings showed that the average crystallite sizes for the WO_3_ synthesized in the nanocomposite were 7.5 nm, with diffraction peaks of 2θ = 23.28° and 28.2° for the corresponding (110) and (200) planes, respectively. The average crystallite size of the SnO_2_ synthesized in the nanocomposite was estimated to be 3.9 nm, with diffraction peaks at 2θ = 32.10°, with a corresponding (101) plane [37,38].

The chemical compositions and surfaces of the WO_3_, SnO_2_, and WO_3_/SnO_2_ nanocomposites were further investigated using XPS (Figure 2 and Table 1). The survey spectrum for the WO_3_/SnO_2_ nanocomposite showed the coexistence of W, Sn, O, and C elements, and no other impurity peak was significantly detected (Figure 2a).

The high-resolution W 4f spectrum for WO_3_ was deconvoluted into two peaks (Figure 2b). The peaks at 36.08 eV and 38.21 eV were assigned to W 4f_7/2_ and W 4f_5/2_ binding energies, respectively, corresponding to the W^6+^ oxidation state. The separation between the W 4f_7/2_ and W 4f_5/2_ peaks was 2.13 eV, which was in good agreement with an earlier report [1]. Figure 2c shows the O 1s peak at 531.1 eV, which was assigned to the lattice oxygens of W–O bonds in WO_3_. The peak at 532.01 eV was due to an oxygen deficiency, the peak at 532.7 eV was assigned to chemisorbed oxygen, and the peak at 533.6 eV was attributed to water and hydroxide. Figure 3d shows the high-resolution Sn 3d spectrum of pure SnO_2_, which was deconvoluted into two peaks with binding energies of 487.6 eV and 496.05 eV. These peaks were assigned to the Sn 3d_5/2_ and Sn 3d_3/2_ binding energies, respectively, for the oxidation state Sn^4+^. The splitting between the Sn 3d_5/2_ and Sn 3d_3/2_ peaks was 8.45 eV, which was in good agreement with a previous study [4]. As displayed in (Figure 2e), the O 1s spectrum was deconvoluted into three peaks at 531.5 eV for lattice SnO_2_, chemisorbed oxygen at 532.7 eV, and water and hydroxide at 534.3 eV. In the WO_3_/SnO_2_ nanocomposite, the W 4f spectrum showed two peaks at 35.74 eV and 38.08 eV, which corresponded to W 4f_7/2_ and W 4f_5/2_ binding energies, respectively, and an oxidation state of W^+6^ (Figure 2f). The spin–orbit separation of W 4f_7/2_ and W 4f_5/2_ was 2.24 eV, which agreed with previous reports [39]. The difference in the W 4f doublet separation for the WO_3_ and WO_3_/SnO_2_ nanocomposites (2.13 eV and 2.24 eV, respectively) was also reported in an earlier study [2]: the W 4f_7/2_ peak was shifted by −0.24 eV, and the W 4f_5/2_ peak was shifted by −0.13 eV, which suggested an interaction between WO_3_ and SnO_2_ (Table 1). Moreover, the Sn 3d spectrum of the WO_3_/SnO_2_ composite had two peaks at 487.69 eV and 496.12 eV, which corresponded to the characteristic Sn 3d_5/2_ and Sn 3d_3/2_ binding energies, respectively. The spin–orbit separation between the Sn 3d_5/2_ and Sn 3d_3/2_ peaks was 8.43 eV (Figure 2g), which agreed with earlier reports [40]. Comparing pure SnO_2_ and WO_3_/SnO_2_, the Sn 3d_5/2_ and Sn 3d_3/2_ peaks were slightly shifted by 0.02 eV, which could have been due to an interaction between WO_3_ and SnO_2_. The O 1s spectrum was deconvoluted into four peaks (Figure 2h) at 531.09 eV, 531.76 eV, 532.8 eV, and 534.3 eV. The main peak at 531.76 eV was attributed to lattice oxygens in the metal oxides WO_3_ and SnO_2_ [33,37,40], the peak at 531.09 eV was assigned to oxygen defects or vacancies, the peak at 532.8 eV was related to chemisorbed oxygen species [41], and the peak at 534.3 eV was attributed to water and hydroxides on the surface of the WO_3_/SnO_2_ nanocomposite, as reported previously [6,32,40]. The binding energies of the lattice oxygens and oxygen vacancies in the WO_3_/SnO_2_ nanocomposite were shifted from those of the single components by −0.34 eV and 0.25 eV, respectively, which suggested an interaction between WO_3_ and SnO_2_ in the nanocomposite. The binding energy for absorbed water/hydroxides in the nanocomposite was similar to that of SnO_2_, which suggested that the surface of the nanocomposite was covered by SnO_2_.

The morphologies of the materials were observed by SEM. WO_3_ was composed of aggregated monodisperse microspheres (diameter: ~1 μm) with rough surfaces (Figure 3a,b). The spaces between these microspheres would allow electrolyte ions to diffuse into the porous nanostructure [15,42]. The roughness of the microsphere surface increased the number of active sites available for electrochemical reactions. Moreover, the SnO_2_ exhibited nanorod structures with ~60 nm widths and ~200 nm lengths (Figure 3c,d). The applied potential and the deposition time are crucial for successful formation of SnO_2_ nanorods [43,44]. In this work, SnO_2_ nanorods were synthesized from SnCl_2_ on a cathode (−0.9 V vs. Ag/AgCl), and the formation of SnO_2_ crystals can be attributed to electrochemical deposition of Sn(OH)_2_, followed by air oxidation during calcination, as shown below (Equations (4)–(7)) [45]:(4)$${\mathrm{SnCl}}_{2}+2\mathrm{OH}\;\to \;\mathrm{Sn}(\mathrm{OH}{)}_{2}+2{\mathrm{Cl}}^{-}$$
(5)$$\mathrm{Sn}(\mathrm{OH}{)}_{2}\;\to \;\mathrm{SnO}+H_{2}O$$
(6)$$\mathrm{SnO}+½\;O_{2}\;\to \;{\mathrm{SnO}}_{2}$$
(7)$$\mathrm{Sn}(\mathrm{OH}{)}_{2}+½\;O_{2}+H_{2}O\;\to \;\mathrm{Sn}(\mathrm{OH}{)}_{4}\;\to \;{\mathrm{SnO}}_{2}+2H_{2}O$$

The SEM images in (Figure 3e,f) indicate the surface morphology of the WO_3_/SnO_2_ nanocomposite. Thin and wrinkled films were uniformly formed on the substrate. These films seemed to cover microspheres, which could be the WO_3_ first deposited on the carbon cloth substrate. This morphology suggested that SnO_2_ could not form coarse nanorods on the rough surfaces of WO_3_, so smaller SnO_2_ crystals formed a network on the surface [46]. In the cross-sectional views (Figure 3g,h), the surface of WO_3_ was covered by nanorods of SnO_2_ [47]. These small crystals and the fine nanorods provide a large surface area, which could enhance the electrochemical performance of this nanocomposite [32].

The elemental compositions of the synthesized materials were measured with EDS (Figure 4). In the pure WO_3_ microspheres, the atomic ratio of W to O was 1:3.09, which agreed with the theoretical ratio of 1:3. In the SnO_2_ nanorods, the atomic ratio of Sn to O was 1:1.58, which was lower than the theoretical ratio of 1:2, suggesting that unreacted Sn^2+^ provided defects or oxygen vacancies in the crystal lattice. The WO_3_/SnO_2_ nanocomposite consisted of W, Sn, and O, with atomic percentages of W (18.45%), Sn (6.31%), and O (75.24%). The Sn content was significantly lower than the W content, which was supported by the SEM observation (Figure 3e,f). The elemental maps (Figure 4d–g) showed similar patterns for the components, which suggests homogeneous coverage of SnO_2_ on the WO_3_ microspheres.

### 2.2. Electrochemical Measurements

#### 2.2.1. Electrochemical Measurements without a Redox-Active Electrolyte

The electrochemical behaviors of the synthesized materials were analyzed with CV and GCD measurements using the non-redox electrolyte Na_2_SO_4_ (Figure 5 and Figure 6).

Figure 5a shows the CV curves for WO_3_ in the potential window −0.2 to 0.7 V vs. Ag/AgCl with various scan rates of 5–50 mV/s. These CV curves demonstrated nonrectangular shapes and medium oxidation peaks at approximately 0.1 V, which indicated a typical combination of pseudocapacitive behavior and electrical double-layer capacitance [48,49]. The oxidation peak clearly increased as the scan rate was increased, indicating the high reactivity of the WO_3_ electrode [50,51]. The specific capacitance was determined from the CV curves by using (Equation (8)) [9,52,53,54]:(8)$$\mathrm{Ccv}({\mathrm{Fg}}^{-1})=\frac{1}{m\nu \left(Vf-Vi\right)}\int_{V_{i}}^{V_{f}}idV$$
where C_cv_ is the gravimetric specific capacitance (F/g) calculated from the measurement, i is the voltammetric current, m is the mass of the active material (mg), ν is the scan rate (mV/s), V_f_ and V_i_ are the bounds of the potential window. The C_cv_ of the WO_3_ electrode was 240, 180, 150, 120, 98, and 84 F/g at scan rates of 5, 10, 20, 30, 40, and 50 mV^−1^, respectively.

Under the same conditions, the CV curves for SnO_2_ displayed quasi-rectangular shapes, with redox peaks at 0.54–0.69 V (oxidative) and 0.12–0.37 V (reductive), which indicated the more pseudocapacitive nature of the SnO_2_ electrode (Figure 5b). As the scan rate was increased, the CV curves for SnO_2_ exhibited more rectangular shapes, and the difference between the redox peaks increased, which suggested relatively slow redox reactions occurring on the surfaces of the SnO_2_ nanorods. The specific capacitance values for the SnO_2_ electrodes were calculated from the CV curves and were 140, 120, 90, 60, 50, and 45 F/g at scan rates of 5, 10, 20, 30, 40, and 50 mV/s, respectively, as shown in (Figure 5e). The CV curves for the WO_3_/SnO_2_ nanocomposite showed no significant redox peak (Figure 5c), which suggested the complementary work of the two components and a synergistic effect arising from interactions between WO_3_ and SnO_2_ (see Figure 2 and Table 1). This WO_3_/SnO_2_ nanocomposite electrode generated greater areas under the CV curves, which indicated a higher specific capacitance than the single-component electrodes (WO_3_ and SnO_2_) (Figure 5e). The specific capacitances of the WO_3_/SnO_2_ nanocomposite were calculated as 530, 410, 280, 210, 160, and 150 Fg^−1^ at scan rates of 5, 10, 20, 30, 40, and 50 mV s^−1^, respectively. Comparing the CV curves and the specific capacitances of the three electrodes, the WO_3_/SnO_2_ nanocomposite electrode had a higher specific capacitance, especially at low scan rates (Figure 5d,e). This enhanced performance for the nanocomposite electrode could be explained by activation of the SnO_2_ via the synergistic effect and the smaller crystals of SnO_2_ formed on WO_3_ (Figure 4). The small SnO_2_ crystals could have enhanced the surface area of the electrode and shortened the electron pathway from the surface of SnO_2_ to WO_3_ [55].

Figure 6a illustrates the GCD curves of the WO_3_ electrode in the potential window—0.2 to 0.7 V vs. Ag/AgCl at various current densities. The nonlinear nature of the charge/discharge processes confirmed the pseudocapacitive nature and the high activity at low potential (<0.1 V). The specific capacitance was calculated from the GCD curves and Equation (9) [9,53,54]:(9)$$C_{G}({\mathrm{Fg}}^{-1})=\mathrm{Ig}\int dt/m\Delta V\left(t\right)$$
where C_G_ is the gravimetric capacitance (F/g) from the GCD measurement, I_g_ is the given current, dt is the discharging time (s), and ΔV(t) is the potential window as a function of t. The C_G_ of WO_3_ electrode was calculated to be 120 Fg^−1^ at a current density of 1.25 Ag^−1^. The SnO_2_ electrode exhibited two steps in the charge process and one step in the discharge process (Figure 6b). At 1.25 Ag^−1^, the specific capacitance of the SnO_2_ electrode was 54 Fg^−1^. Figure 6c shows the GCD curves for the WO_3_/SnO_2_ nanocomposite electrode at different current densities. The shapes of the GCD curves were more triangular, which indicated that the nanocomposite electrode exhibited better pseudocapacitive nature than the others, as was also suggested by the CV measurements (Figure 6). The specific capacitance of the WO_3_/SnO_2_ nanocomposite was determined to be 180, 120, 80, and 68 Fg^−1^ at the various current densities 1.25, 1.88, 2.82, and 5.00 Ag^−1^, respectively. Figure 6d provides a comparison of the GCD curves for the SnO_2_, WO_3_, and WO_3_/SnO_2_ electrodes at the same current density of 1.25 Ag^−1^. Figure 6e shows the EIS results for the WO_3_, SnO_2_, and WO_3_/SnO_2_ nanocomposite electrodes, and the magnified EIS data for the electrode are shown in (Figure 6f). The charge transfer resistance values (Rct) calculated for the WO_3_, SnO_2_, and WO_3_/SnO_2_ nanocomposite electrodes were 2.12 Ω, 1.94 Ω, and 1.72 Ω, respectively. The lower Rct of the WO_3_/SnO_2_ nanocomposite electrode indicated rapid charge transfer in the nanocomposite. The resistance of the electrode–electrolyte interference (Rs), which corresponds to the *x*-axis intercept in the high-frequency region, was calculated to be 2.37 Ω, 2.32 Ω, and 1.87 Ω for the WO_3_, SnO_2_, and WO_3_/SnO_2_ nanocomposite electrodes, respectively. The low Rs suggested that the WO_3_/SnO_2_ nanocomposite contacted the electrolytes with large surface areas because of its small grain sizes (Figure 4).

#### 2.2.2. Effects of Redox-Active Electrolyte

Electrochemical analyses of the WO_3_, SnO_2_, and WO_3_/SnO_2_ electrodes were performed in the presence of the redox additive 0.2 M K_3_Fe (CN)_6_ in the 1 M Na_2_SO_4_ aqueous electrolyte. First, CV measurements were conducted with the carbon cloth, WO_3_, SnO_2_, and WO_3_/SnO_2_ nanocomposite electrodes (Figure 7a).

While the carbon cloth was almost inactive, the CV curves for the metal oxide electrodes exhibited quasi-rectangular shapes, with a pair of redox peaks from [Fe (CN)_6_]^3−^/[Fe (CN)_6_]^4−^ at 0.35–0.43 V and 0.18–0.26 V. The WO_3_/SnO_2_ nanocomposite electrode had the largest area at a scan rate of 50 mV/s, which indicated the highest specific capacitance. The WO_3_/SnO_2_ nanocomposite electrode also resulted in higher intensities of redox peaks than the other electrodes, which indicated that this electrode exhibited the highest activity for the redox reactions of [Fe(CN)_6_]^3−^/[Fe(CN)_6_]^4−^. Moreover, considering the high peak intensities, the difference between the oxidation and reduction potentials was smaller than those of the other electrodes. The reduction peak for [Fe(CN)_6_]^3−^ with the WO_3_/SnO_2_ nanocomposite electrode was at the highest potential among the electrodes, while the oxidation potential for [Fe(CN)_6_]^4−^ with the WO_3_/SnO_2_ nanocomposite electrode was higher than the others. These behaviors suggested that the WO_3_/SnO_2_ nanocomposite catalyzed the redox reactions of [Fe(CN)_6_]^3−^/[Fe(CN)_6_]^4−^, which is preferable for energy storage because of the low overpotential. The catalytic activity of the WO_3_/SnO_2_ nanocomposite could be due to interactions between WO_3_ and SnO_2_, as suggested by the XPS measurements (Figure 2 and Table 1). Therefore, the CV curves generated with two sets of conditions were compared, i.e., with K_3_Fe (CN)_6_ and without K_3_Fe (CN)_6_ (Figure 7b). With K_3_Fe (CN)_6_, the areas of the CV curves were larger, which indicated that the redox additive increased the charge storage capacity. Using the CV curves generated at different scan rates (Figure 7c–e), the specific capacitance of the electrode in the redox-active electrolyte was calculated. At scan rates of 5, 10, 20, 30, 40, and 50 mV/s, the respective specific capacitance values were 440, 360, 280, 240, 190, and 150 F/g for the WO_3_ electrode; 310, 220, 150, 130, 120, and 110 F/g for the SnO_2_ electrode; and 640, 520, 340, 330, 310, and 290 F/g for the WO_3_/SnO_2_ nanocomposite electrode. Moreover, the addition of K_3_Fe(CN)_6_ resulted in more pseudocapacitive (more rectangular CV curves) for the WO_3_ and WO_3_/SnO_2_ nanocomposite electrodes. In details, weak and broad reduction bands newly appeared at −0.1–0.0 V for the WO_3_ electrode and 0.0–0.1 V for the WO_3_/SnO_2_ electrode in the presence of K_3_Fe(CN)_6_ (Figure 7b,c,e). The pseudocapacitive behavior suggested that K_3_Fe(CN)_6_ provided the additional redox reactions of the [Fe(CN)_6_]^3−^/[Fe(CN)_6_]^4−^ couple to increase the specific capacitance of these electrodes [56,57], and also promoted the redox reactions of the WO_3_ component. That is, these metal oxides (WO_3_ and WO_3_/SnO_2_ nanocomposites) transferred electrons to the [Fe(CN)_6_]^3−^/[Fe(CN)_6_]^4−^ pair in the electrolyte solution via their own redox reactions. The plots of the specific capacitance for various electrodes against the different scan rates showed that only the WO_3_/SnO_2_ nanocomposite electrode exhibited a larger enhancement of the specific capacitance by K_3_Fe(CN)_6_ at the high scan rate versus the low scan rate (Figure 7f). This enhancement at the high scan rate was due to the catalytic activity of the WO_3_/SnO_2_ nanocomposite in facilitating the redox reactions of [Fe(CN)_6_]^3−^/[Fe(CN)_6_]^4−^, as mentioned above.

The kinetics of the energy storage process were analyzed from the perspectives of peak current and scan rate (Figure 7g). In general, the relationship between the peak current and the scan rate can be described as in (Equation (10)) [58,59]:I_p_ = aν^b^(10)
where I_p_ is the peak oxidation current (mA), ν is the scan rate (mV/s), and both a and b are adjustable parameters. The b value is obtained from the slope of a plot of log I_p_ versus log ν. When the charge storage process is a surface-controlled capacitive process (ion adsorption/desorption), the value of b is 1. On the other hand, if diffusion is the rate-controlling process, b is 0.5 [60]. With and without the K_3_Fe(CN)_6_ electrolyte, the b values of the WO_3_/SnO_2_ nanocomposite electrode were 0.55 and 0.71, respectively. These b values suggested that addition of the K_3_Fe(CN)_6_ electrolyte favored diffusion control of the energy storage process because the approach of the redox-active species enabled charge transfer at the electrode surface. Moreover, the b values for the WO_3_ and SnO_2_ electrodes were 0.62 and 0.75, respectively. These b values were higher than that of the WO_3_/SnO_2_ nanocomposite electrode, which indicated that these single-component electrodes were less affected by diffusion of the redox-active species.

The GCD measurements made in the presence of the K_3_Fe(CN)_6_ electrolyte were not adequate to calculate the specific capacitance values of the synthesized electrodes because the discharge process took longer than the charge time (see Appendix A). This extension of the discharge process could be due to reduction of [Fe(CN)_6_]^3−^ to [Fe(CN)_6_]^4−^, which could be used for the redox-flow battery system. Therefore, GCD measurements were conducted for stability testing (Figure 7h). The coulombic efficiency of the electrode was calculated with Equation (11):η = t_d_/t_c_ × 100(11)
where η is the coulombic efficiency, t_d_ is the discharging time, and t_c_ is the charging time (s) [9]. After 2000 GCD cycles at a current density of 2.5 A/g in the presence of K_3_Fe(CN)_6_ electrolyte, the WO_3_/SnO_2_ nanocomposite electrode demonstrated good cycling stability, capacitance retention of 94.7%, and a coulombic efficiency of 99.9%. The inset graph presents the GCD curves for the first 10 and the last 10 cycles, and the shapes of the curves remained triangular, which indicated the pseudocapacitive nature and good capacitive behavior even after 2000 cycles. For comparison, the WO_3_ electrode exhibited lower capacitance retention (90.4%) under the same conditions. This difference indicated that K_3_Fe(CN)_6_ improved the stability of the electrode, not just the pseudocapacitor. EIS analyses of the WO_3_, SnO_2_, and WO_3_/SnO_2_ nanocomposite electrodes were carried out to determine the effects of K_3_Fe (CN)_6_ on the conductivities of these electrodes. In the electrolyte system containing 1 M Na_2_SO_4_ with 0.02 M K_3_Fe(CN)_6_, the R_ct_ values of the WO_3_, SnO_2_, and WO_3_/SnO_2_ nanocomposite electrodes were 1.71 Ω, 1.69 Ω, and 1.56 Ω, respectively, and the Rs values of these electrodes were 2.78 Ω, 2.13 Ω, and 1.65 Ω, respectively (Table 2).

The WO_3_/SnO_2_ nanocomposite electrode exhibited a lower Rct than the other electrodes in the electrolyte systems with/without K_3_Fe(CN)_6_. For all electrodes used in this study, K_3_Fe(CN)_6_ drastically decreased R_ct_, which indicated that K_3_Fe(CN)_6_ facilitated charge transfer between the active materials. That is, K_3_Fe(CN)_6_ activated the electrical behaviors of the metal oxides used in this study. The R_s_ values also showed improvement, except for that of the WO_3_ electrode, which was attributed to an additional redox reaction occurring between the electrode surface and the electrolyte, in addition to the increased ion concentration of the electrolyte solution.

#### 2.2.3. Electrochemical Performance of Symmetric Systems

For practical application of the WO_3_/SnO_2_ nanocomposite electrode, a symmetric supercapacitor system with identical electrodes was constructed with an aqueous electrolyte containing 1 M NaSO_4_ and 0.02 M K_3_Fe(CN)_6_. The total mass of the WO_3_/SnO_2_ nanocomposite on both electrodes was ~3.4 mg (1.7 mg for each). To confirm the optimal voltage window of the device, CV curves were measured over different potential ranges (from 1.0 to 1.8 V) at a scan rate of 50 mVs^−1^, as shown in (Figure 8a). The CV profile showed that the working potential was extended to 1.8 V, and water hydrolysis occurred at 1.9 V. As the potential range was increased, the shapes of the CV curves changed from quasi-rectangular (up to 1.4 V) to rectangular with redox peaks (higher than 1.6 V). This change indicated that a potential range larger than 1.6 V converted the pseudocapacitive mechanism to an energy storage mechanism, including the redox reactions of [Fe(CN)_6_]^3−^/[Fe(CN)_6_]^4−^ ions.

The CV curves of the symmetric WO_3_/SnO_2_ nanocomposite electrode systems with and without K_3_Fe(CN)_6_ showed nearly rectangular shapes, which indicated pseudocapacitive behavior, even at a high scan rate of 50 mV/s (Figure 8b,c). Comparing the two systems, the CV profile of the WO_3_/SnO_2_ nanocomposite electrode with K_3_Fe(CN)_6_ had a greater area than that without K_3_Fe(CN)_6_, as expected from the half-cell experiment above (Figure 7b). GCD measurements of the symmetric WO_3_/SnO_2_ nanocomposite electrodes with and without K_3_Fe (CN)_6_ were also performed with various current densities (Figure 8e,f), and the GCD curves generated at 1.25 A/g were compared (Figure 8g). The IR drop in the initial phase of discharge was decreased in the presence of K_3_Fe(CN)_6_ (dropped from 1.8 V to ~1.0 V without K_3_Fe(CN)_6_ and from 1.8 V to ~1.3 V with K_3_Fe(CN)_6_). The specific capacitance values were determined at a current density of 1.25 A/g and were 285 F/g with K_3_Fe(CN)_6_ and 186 F/g without K_3_Fe(CN)_6_. The coulombic efficiencies were low in these cases, which could be explained by a minor water hydrolysis and the diffusion of [Fe(CN)_6_]^3−^/[Fe (CN)_6_]^4−^ ions. At higher potential than 1.6 V, the edge of water electrolysis started (Figure 8a), and the minor gas generation resulted in the energy loss. Using K_3_Fe(CN)_6_, the reaction products ([Fe(CN)_6_]^3−^ and [Fe(CN)_6_]^4−^, mutually) diffused into the whole electrolyte solution in the cell, which decreased the availability of these ions for the discharge process [28].

The specific energy density (SE) and the specific power density (SP) of the WO_3_/SnO_2_ symmetric supercapacitor system with/without redox activity were important parameters for determining the feasibility of the device application. S_E_ and S_P_ were calculated using Equations (12) and (13) [53,54]:(12)$$S_{E}({\mathrm{Whkg}}^{-1})=\frac{I}{M\left(3600\right)}\int_{0}^{t_{d}}Vdt$$
(13)$$S_{p}(W\;{\mathrm{Kg}}^{-1})=\frac{ED\;X\;3600}{t_{d}}$$
where M is the total mass of both the positive and negative electrodes, I is the current, and V is the potential as a function of discharge time (t_d_). The Ragone diagram for the energy density vs. power density of the WO_3_/SnO_2_ nanocomposite electrodes with/without additive is shown in (Figure 8h). The WO_3_/SnO_2_ symmetric supercapacitor system with the K_3_Fe(CN)_6_ electrolyte exhibited the highest S_E_ of 64 Whkg^−1^ at a S_P_ of 542 Wkg^−1^, whereas without K_3_Fe(CN)_6_, the S_E_ was 35 Whkg^−1^ for a S_P_ of 468 Wkg^−1^ (in both cases, the current density was 1.25 A/g). Compared with previous reports, the symmetric device with the WO_3_/SnO_2_ nanocomposite electrodes exhibited excellent electrochemical performance in the Na_2_SO_4_/K_3_Fe(CN)_6_ electrolyte (Table 3). In the symmetric system, the active materials simultaneously worked as positive (anode) and negative (cathode) electrodes, and the GCD curves confirmed the reversible balanced redox reactions occurring at the anode and cathode [61,62]. Although the WO_3_/SnO_2_ ratio and the concentration of K_3_Fe(CN)_6_ should be optimized in future studies, the approach in this study would be useful for designing metal oxide nanocomposites for supercapacitors.

## 3. Discussion

In this study, microspheres of WO_3_, nanorods of SnO_2_, and WO_3_/SnO_2_ nanocomposite thin films were synthesized via chemical bath deposition, electrodeposition, and calcination. The hexagonal phase of WO_3_ and the tetragonal phase of SnO_2_ were observed in both the single-component and nanocomposite materials. The average WO_3_ crystal size in the nanocomposite was smaller than that in single-component WO_3_. An interaction between WO_3_ and the SnO_2_ in the nanocomposite was also suggested by the shifts in binding energies measured by XPS. SEM observations indicated the mesoporous surface structures of these nanomaterials, which are preferable for supercapacitor applications.

The CV measurements revealed that the specific capacitance of the WO_3_/SnO_2_ nanocomposite electrode reached 530 Fg^−1^ without K_3_Fe(CN)_6_ and 640 Fg^−1^ with K_3_Fe(CN)_6_ at a low scan rate of 5 mV/s. The improved capacitance in the presence of K_3_Fe(CN)_6_ was explained by the redox reactions of [Fe(CN)_6_]^3−^/[Fe(CN)_6_]^4−^ ions, in addition to the redox reactions of the WO_3_/SnO_2_ nanocomposite. Moreover, the WO_3_/SnO_2_ nanocomposite and K_3_Fe(CN)_6_ could mutually facilitate their redox reactions. The improved specific capacitance was also observed for single-component electrodes (both of WO_3_ and SnO_2_). Therefore, this approach can be used for other electrodes using these metal oxides, although the reaction efficiency could be differed by the electrode species.

Using the symmetric configurations of WO_3_/SnO_2_ nanocomposite electrodes, the addition of K_3_Fe(CN)_6_ to the 1 M Na_2_SO_4_ electrolyte generated a high energy density and a power density of 64 Whkg^−1^ at 542 Wkg^−1^, respectively. The capacitive retention efficiencies for the WO_3_/SnO_2_ and WO_3_ electrodes with the K_3_Fe(CN)_6_ electrolyte was 94.7% and 90.4%, respectively, as indicated by GCD curves after 2000 cycles at 2.5 Ag^−1^. The higher cycling stability and the excellent coulombic efficiency (99.9%) of the WO_3_/SnO_2_ nanocomposite electrode relative to those of the pure WO_3_ electrode were due to the high catalytic activity of the WO_3_/SnO_2_ nanocomposite for the redox reactions of [Fe(CN)_6_]^3−^/[Fe(CN)_6_]^4−^ ions. As a result, the binder-free WO_3_/SnO_2_ nanocomposite with a redox-active electrolyte constituted a promising system for pseudocapacitive energy storage devices. Although the optimal configuration of the WO_3_/SnO_2_ nanocomposite and the electrolyte system should be studied further in the future, this approach can also be used with other metal oxide nanocomposites in reactive electrolytes.

## 4. Materials and Methods

Sodium tungstate dihydrate (Na_2_WO_4_·2H_2_O) was purchased from Alfa Aesar India, and tin (II) chloride hydrate (SnCl_2_·2H_2_O) was purchased from Alfa Aesar United States. Hydrochloric acid (HCl, ~35% *v/v*) and ammonium sulfate ((NH_4_)_2_SO_4_) were purchased from Duksan. Carbon cloth (WOS1009) was obtained from CeTech Co., Ltd., and sodium sulfate (Na_2_SO_4_, anhydrous, ≥99%) was from Honeywell/Fluka. Potassium ferricyanide (K_3_Fe(CN)_6_, 99%) was purchased from Acros Organics. All of these chemicals were used without further purification. Aqueous solutions were freshly prepared with ultrapure water (resistivity of 18.2 MΩcm; Yamato, Japan) throughout the experiments. Characterization was performed over the range 2θ = 10° to 60° at room temperature with an X-ray diffractometer (XRD, Bruker, D2 phaser Karlsruhe, German) using a copper radiation source (λ = 0.154 nm), operating at 40 kV and 30 mA, with a step size of 0.05 and step time of 5 sec.; X-ray photoelectron spectroscopy (XPS, VG scientific ESCALAB 250, Birmingham, UK); and field emission scanning electron microscopy (FE-SEM, at 15 kV acceleration voltage) equipped with an energy-dispersive X-ray spectroscopy (EDS) analyzer (JSM-6500F, JEOL, Tokyo, Japan).

### 4.1. Chemical Bath Deposition of WO_3_ Nanosphere Thin Films

WO_3_ nanospheres were deposited on a carbon cloth substrate by the CBD method. First, a carbon cloth substrate (1 × 1 cm^2^) was treated with 1 M HNO_3_, acetone, ethanol, and water sequentially with ultrasonication for 20 min and then dried overnight in a vacuum. In a typical preparation of WO_3_ nanospheres, Na_2_WO_4_·H_2_O (0.82 g) was dissolved in 50 mL of water. After stirring for 20 min, 1 M HCl was added dropwise to produce a pH of 1.2 ± 1.0. After stirring for 30 min, (NH_4_)_2_SO_4_ (1.32 g) was added to the above solution. After stirring for 10 min, 40 mL of the above mixture was transferred to a 100 mL beaker, and the carbon cloth substrate was immersed in it. Then, the beaker was heated on a hotplate at 80 °C for 4 h. The carbon cloth was removed from the bath, rinsed several times with water, and dried overnight at room temperature. Finally, the film was calcined at 400 °C for 2 h. The mass of WO_3_ loaded onto the carbon cloth substrate was measured by subtracting the weight of the original carbon cloth substrate from the weight of the product and determined to be 1.2 mg.

### 4.2. Electrochemical Deposition of SnO_2_ Nanorods

SnO_2_ nanorods were synthesized through electrochemical deposition. First, a carbon cloth substrate with an area of 1 × 1 cm^2^ was treated with 1 M HNO_3_, acetone, ethanol, and water through ultrasonication and sequentially dried overnight at ambient temperature. Next, an aqueous solution of SnCl_2_ (0.112 g) was prepared in 50 mL of water, and 2.6 mL of HCl was added. After stirring for 30 min, 25 mL of the solution was transferred to a 100 mL beaker, and a three-electrode system was set up: carbon cloth was used as the working electrode (WE), Ag/AgCl in 3 M Cl was used as the reference electrode (RE), and a platinum wire was used as the counter electrode (CE). Then, in a solution of SnCl_2_ with HCl, potentiostatic electrodeposition was performed at an applied potential of −0.9 V for 10 min using a galvanostat/potentiostat (Hokuto-Denko, Model HA-151B) at ambient temperature. After deposition, the WE was rinsed with water and dried overnight at ambient temperature. Finally, the deposited material on the carbon cloth was calcined at 400 °C for 2 h. The deposited mass was measured by subtracting the weight of the original carbon cloth substrate from the weight of the product and determined to be 0.4 mg.

### 4.3. Synthesis of a WO_3_/SnO_2_ Nanocomposite Electrode

WO_3_/SnO_2_ nanocomposite thin films were synthesized through CBD and the electrochemical deposition method. First, the carbon cloth substrate of area (1 × 1 cm^2^) was sequentially treated by ultrasonication in 1 M HNO_3_, acetone, ethanol, and ultrapure water and then dried overnight at ambient temperature. WO_3_ microspheres were deposited on the carbon cloth by CBD, and then the carbon cloth/WO_3_ was immersed in the SnCl_2_/HCl solution (25 mL, 0.112 g, and 0.65 M) in a 100 mL beaker [63]. Then, the three-electrode system was set up: the WE were carbon cloth supported by WO_3_ nanospheres, the RE was Ag/AgCl in 3 M Cl, and the CE was a platinum wire. Then, potentiostatic electrodeposition was performed at an applied potential of −0.9 V for 10 min using a galvanostat/potentiostat (Hokuto-Denko, Model HA-151B) at ambient temperature. After deposition, the WE were rinsed with water and then dried overnight at ambient temperature. Finally, the deposited material on the carbon cloth was calcined at 400 °C for 2 h on a hot plate. The deposited mass was measured by subtracting the weight of the original carbon cloth substrate from the weight of the product and was determined to be 1.6 mg.

### 4.4. Electrochemical Analyses

All electrochemical analyses were performed using a standard three-electrode system in an electrochemical cell connected to an electrochemical workstation (ZAHNER mass system, model xpot-26366, Germany). An Ag/AgCl electrode in 3 M KCl was used as the RE; a Pt wire as CE; and the WO_3_, SnO_2_, and WO_3_/SnO_2_ nanocomposites formed on carbon cloth were used as WEs. An aqueous solution of 1 M Na_2_SO_4_ with/without 0.02 M K_3_Fe (CN)_6_ was used as the electrolyte for all electrochemical measurements. The electrochemical performance of WO_3_ and SnO_2_, as well as the WO_3_/SnO_2_ nanocomposite electrodes, was studied using cyclic voltammetry (CV) over the range −0.2 V to 0.7 V vs. Ag/AgCl at different scan rates and galvanostatic charge-discharge (GCD) at different current densities. Electrochemical impendence spectroscopy (EIS) was performed with an amplitude of 10 mV over the frequency range 100 kHz to 100 mHz in an open circuit and described with a Nyquist plot.

## 5. Conclusions

In this work, we developed a preparation method of WO_3_/SnO_2_ nanocomposite on carbon cloth for supercapacitor electrodes. WO_3_ nanospheres were first deposited on a carbon cloth substrate by a chemical bath deposition method, and then a SnO_2_ layer was formed by an electrochemical deposition and calcination. The WO_3_/SnO_2_ nanocomposite on carbon cloth exhibited a higher specific capacitance (530 F/g at 5 mV/s scan rate) than that of single-component electrodes (240 F/g for WO_3_ and 140 F/g for SnO_2_) in an aqueous electrolyte of 1M Na_2_SO_4_. The addition of 0.02 M K_3_Fe(CN)_6_ redox-active electrolyte into the 1M Na_2_SO_4_ electrolyte further improved the charge storage performance of electrodes (440 F/g for WO_3_, 310 F/g for SnO_2_, and 640 F/g for WO_3_/SnO_2_ nanocomposite). These results show the synergistic effects of WO_3_/SnO_2_ nanocomposites and the advantages of using redox-active electrolytes for supercapacitor systems. Using a symmetric configuration of WO_3_/SnO_2_ nanocomposite electrodes, the electrolyte solution of 0.02 MK_3_Fe(CN)_6_ and 1 M Na_2_SO_4_ electrolyte demonstrated a high energy density and a power density of 64 Whkg^−1^ at 542 Wkg^−1^, respectively. The capacitive retention efficiency for the WO_3_/SnO_2_ and WO_3_ electrodes with the K_3_Fe(CN)_6_ electrolyte was 94.7% after 2000 cycles at 2.5 Ag^−1^. Thus, the WO_3_/SnO_2_ nanocomposite electrode grown on carbon cloth is a promising candidates as a binder-free electrode for the high performance of a supercapacitor device, and the supercapacitor system using redox-active K_3_Fe(CN)_6_ electrolyte was proposed. These strategies developed in this study can also be applied to other metal oxide nanocomposites to improve the performance of supercapacitors.

## Figures and Tables

**Figure 1 ijms-24-06045-f001:**
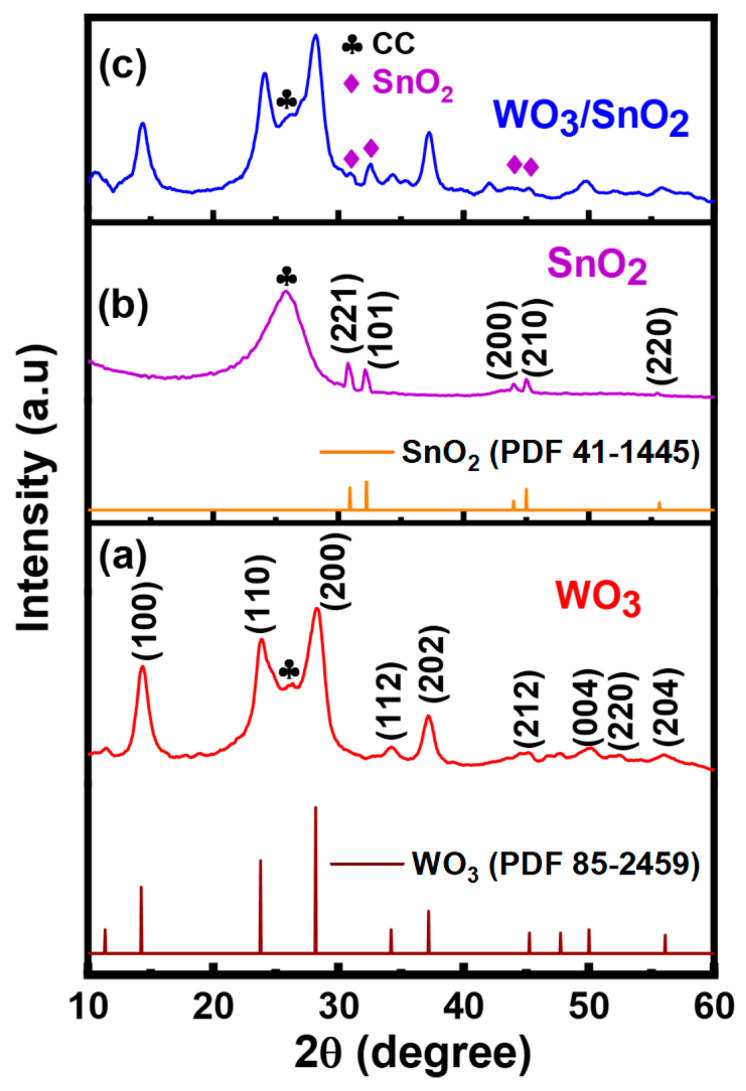
XRD patterns for (**a**) WO_3_, (**b**) SnO_2_, and (**c**) WO_3_/SnO_2_ nanocomposites. The clubs (♣) and diamonds (♦) represent peaks of CC and SnO_2_, respectively.

**Figure 2 ijms-24-06045-f002:**
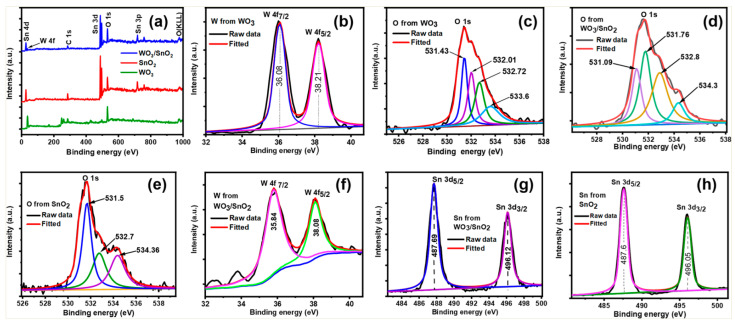
XPS data for WO_3_, SnO_2_, and the WO_3_/SnO_2_ nanocomposite: (**a**) survey spectrum of the WO_3_/SnO_2_ nanocomposite, high-resolution (**b**) W 4f spectrum of WO_3_, (**c**) O 1s spectrum of WO_3_, (**d**) Sn 3d spectrum of SnO_2_, (**e**) O 1s spectrum of SnO_2_, (**f**) W 4f spectrum of WO_3_/SnO_2_, (**g**) Sn 3d spectrum of WO_3_/SnO_2_, and (**h**) O 1s spectrum of the WO_3_/SnO_2_ nanocomposite.

**Figure 3 ijms-24-06045-f003:**
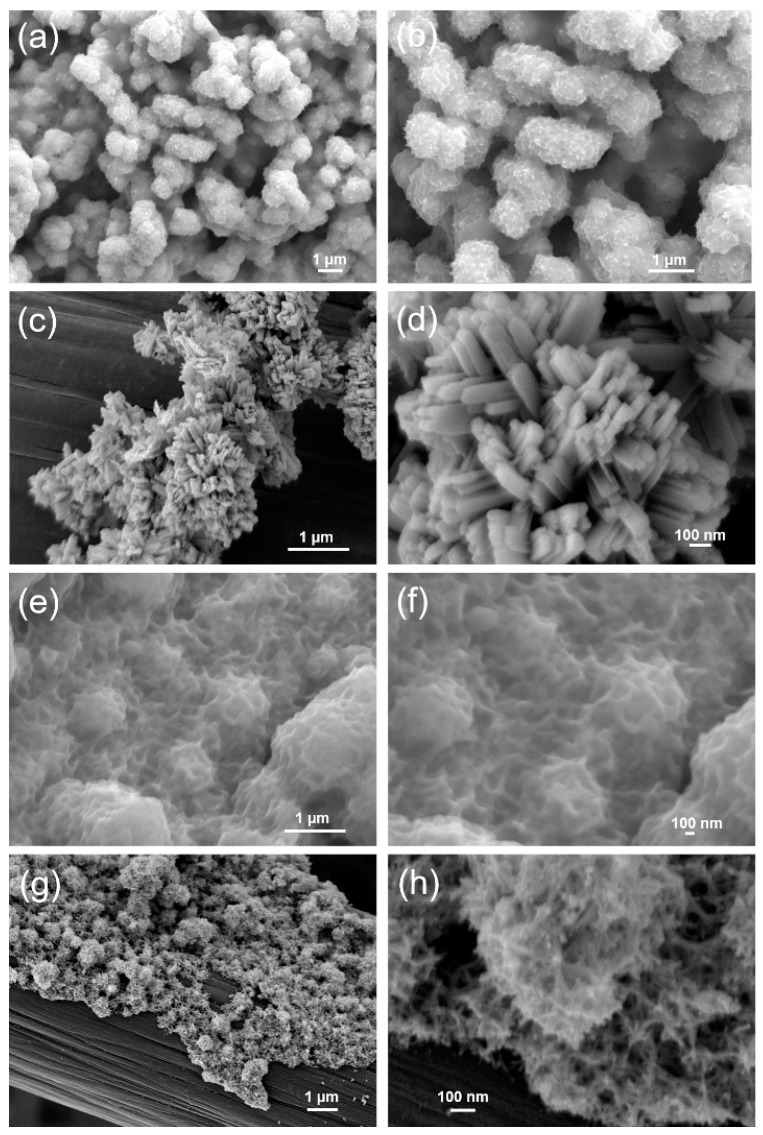
FE-SEM images of (**a**,**b**) WO_3_ grown on a carbon cloth, (**c**,**d**) SnO_2_ grown on a carbon cloth, (**e**,**f**) WO_3_/SnO_2_ nanocomposite, and (**g**,**h**) cross-sectional image of WO_3_/SnO_2_ nanocomposite at low and high magnification, respectively.

**Figure 4 ijms-24-06045-f004:**
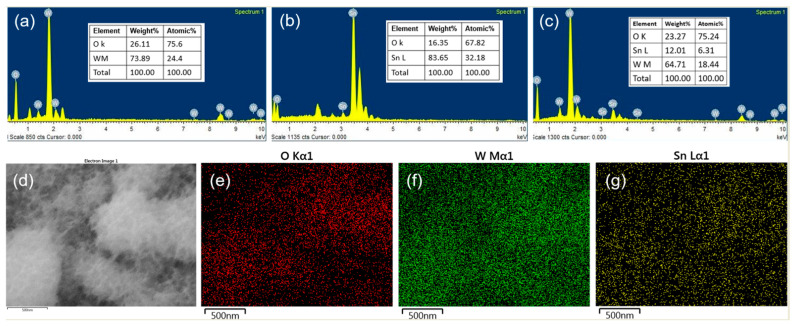
EDS spectra of synthesized (**a**) WO_3_ microspheres, (**b**) SnO_2_ nanorods, and (**c**) WO_3_/SnO_2_ nano-composites; (**d**) SEM image and elemental maps of WO_3_/SnO_2_ nanocomposites for (**e**) O, (**f**) W, and (**g**) Sn.

**Figure 5 ijms-24-06045-f005:**
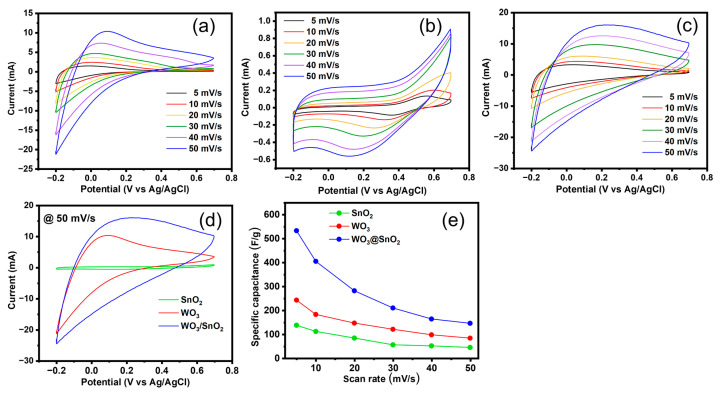
CV curves for (**a**) WO_3_, (**b**) SnO_2_, and (**c**) WO_3_/SnO_2_ nanocomposite electrodes at various scan rates from 5 mV/s to 50 mV/s with 1 M Na_2_SO_4_ electrolyte; (**d**) comparison of CV curves for WO_3_, SnO_2_, and WO_3_/SnO_2_ at 50 mV/s; (**e**) specific capacitance vs. scan rate for WO_3,_ SnO_2_, and WO_3_/SnO_2_ from CV curves.

**Figure 6 ijms-24-06045-f006:**
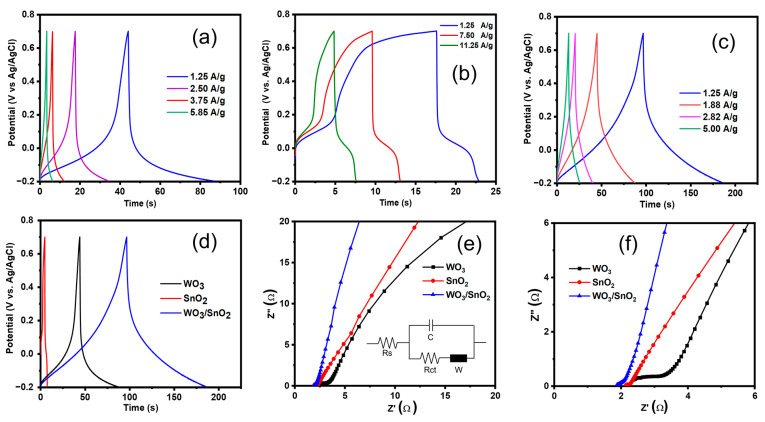
GCD curves determined for electrodes with 1 M Na_2_SO_4_ electrolyte and at different current densities: (**a**) WO_3_, (**b**) SnO_2_, and (**c**) WO_3_/SnO_2_; (**d**) comparison of GCD curves for WO_3_, SnO_2_, and WO_3_/SnO_2_ with the same current density of 1.25 A/g; (**e**) EIS for WO_3_, SnO_2_, and WO_3_/SnO_2_ nanocomposite electrodes; and (**f**) magnified EIS for the inset in (**e**), and the inset in (**e**) is the model circuit used for the analysis.

**Figure 7 ijms-24-06045-f007:**
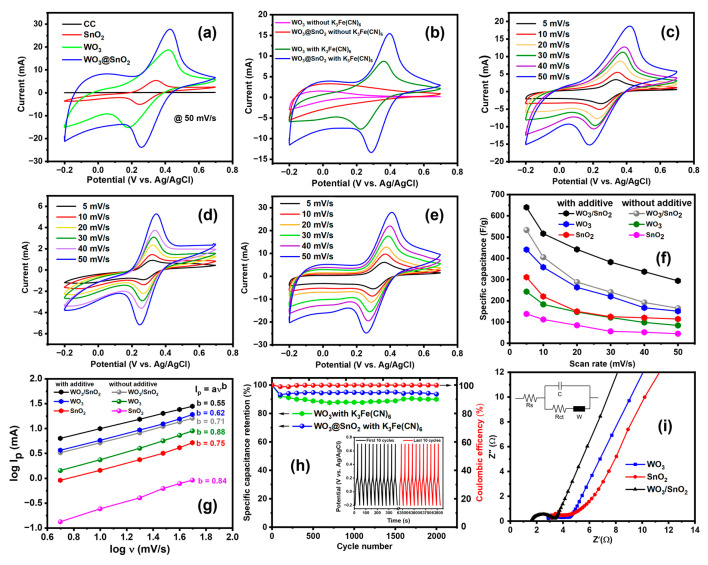
CV curves for (**a**) various electrodes at scan rate of 50 mV/s with K_3_Fe(CN)_6_ and (**b**) WO_3_ and the WO_3_/SnO_2_ nanocomposite electrodes with/without K_3_Fe(CN)_6_ at scan rate of 30 mV/s; CV curves of (**c**) WO_3_ electrode, (**d**) SnO_2_ electrode, and (**e**) WO_3_/SnO_2_ nanocomposite electrode with K_3_Fe(CN)_6_ at various scan rates; (**f**) specific capacitance of various electrodes with/without K_3_Fe(CN)_6_ at various scan rates; (**g**) double logarithmic plots of scan rate vs. peak current for WO_3_, SnO_2_, and the WO_3_/SnO_2_ nanocomposite; (**h**) cycling stability and coulombic efficiency versus cycle number for the WO_3_/SnO_2_ nanocomposite electrode with K_3_Fe(CN)_6_; (**i**) EIS analyses of WO_3_, SnO_2_, and WO_3_/SnO_2_ nanocomposite electrodes with K_3_Fe(CN)_6_ over the frequency range 100 kHz to 100 mHz.

**Figure 8 ijms-24-06045-f008:**
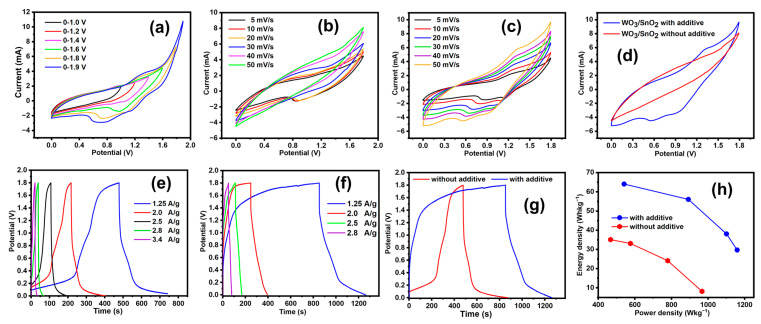
(**a**) CV curves of the WO_3_/SnO_2_ symmetric electrode with different applied potential windows at 50 mV/s, symmetric CV curves of the WO_3_/SnO_2_ symmetric electrode at different scan rates (**b**) without the redox-active electrolyte and (**c**) with the redox-active electrolyte; (**d**) comparison of the CV curves for the WO_3_/SnO_2_ symmetric device with and without the redox-active electrolyte at 50 mV/s; GCD curve for the WO_3_/SnO_2_ symmetric electrode at different current densities (**e**) without the redox-active electrolyte and (**f**) with the redox-active electrolyte; (**g**) comparison of the GCD curve-based symmetric electrode at 1.25 A/g; (**h**) Ragone plots of the WO_3_/SnO_2_ symmetric device.

**Table 1 ijms-24-06045-t001:** Binding energies (eV) for W, Sn, and O in WO_3_, SnO_2_, and WO_3_/SnO_2_ nanocomposites determined by deconvolution of XPS peaks.

		W 4f_7/2_	W 4f_5/2_	Doublet Separation
WO_3_		36.08	38.21	2.13
WO_3_/SnO_2_		35.84	38.08	2.24
		Sn 3d_5/2_	Sn 3d_3/2_	
SnO_2_		487.60	496.05	8.45
WO_3_/SnO_2_		487.69	496.12	8.43
	O in Lattice Metal Oxide	O Deficiency (Defects)	Chemisorbed O	O in Water, Hydroxide (OH^−^/H_2_O)
WO_3_	531.43	532.01	532.72	533.60
SnO_2_	531.50	-	532.70	534.38
WO_3_/SnO_2_	531.09	531.76	532.80	534.30

**Table 2 ijms-24-06045-t002:** Charge Transfer Resistance (R_ct_) and Electrochemical Resistance (R_s_) of Different Electrodes with/without K_3_Fe(CN)_6_.

Electrode	With K_3_Fe(CN)_6_		Without K_3_Fe(CN)_6_	
	Rs (Ω)	Rct (Ω)	Rs (Ω)	Rct (Ω)
WO_3_	2.78	1.71	2.37	2.12
SnO_2_	2.13	1.69	2.32	1.94
WO_3_/SnO_2_	1.65	1.56	1.87	1.72

**Table 3 ijms-24-06045-t003:** Comparison of the electrochemical performance for WO_3_-based active electrode materials with the parameters for supercapacitor applications.

**Electrode**	**Specific Capacitance** **(Fg^−1^)**	**Current Density** **(Ag^−1^)**	**Energy Density** **(Whkg^−1^)**	**Power Density** **(Wkg^−1^)**	**Stability** **(Retention, Cycles)**	**Electrolyte**	**Ref.**
MnO_2_-WO_3_	657@5 mVs^−1^	NA	10.8	650	92%, 2000	Na_2_SO_4_	[21]
RuO_2_-WO_3_	NA	NA	16.92	540	NA	NA	[12]
WO_3_-MnO_2_/WO_3_	88.6	4 mAcm^−2^	24.13	915	95%, 2500	CMC-Na_2_SO_4_	[14]
WO_3_-RGO	495	1	NA	NA	85%, 1000	0.5 M H_2_SO_4_	[13]
WO_3_@CuO	248.2	1	NA	NA	85.2%, 1500	6 M KOH	[60]
WO_3_/Se(ASC)	0.858	0.2	0.047	0.345	74%, 4000	PVA-H_2_SO_4_	[2]
WO_3_/SnO_2_	530@5 mVs^−1^	NA	35	468	NA	1 M Na_2_SO_4_	This work
WO_3_/SnO_2_	640@5 mVs^−1^	NA	64	542	94.7%, 2000	0.02 M K_3_Fe(CN)_6_/1 M Na_2_SO_4_	This work

## Data Availability

The data presented in this study are available on request from the corresponding author.

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
