# Peer review of "Enhanced Performance of WO3/SnO2 Nanocomposite Electrodes with Redox-Active Electrolytes for Supercapacitors"

_ijms, 2023, doi:10.3390/ijms24076045_

Round 1

Reviewer 1 Report

This manuscript reports the Enhanced Performance of WO3/SnO2 Nanocomposite Electrodes with Redox-Active Electrolytes for Supercapacitors. The electrochemical activities of WO3/SnO2 Nanocomposite Electrodes was investigated in K3Fe(CN)6. K3Fe(CN)6 electrolyte. WO3/SnO2 nanocomposite electrode demonstrated excellent specific capacitance. This manuscript needs following minor revision before publication:

1.      While comparing the electrochemical performance of any electrode, the electrochemical performance should be compared in similar operating conditions. The electrochemical performance of the WO3-based active electrode materials should be compared in redox active electrolytes.

2.      Authors should explain suitable reasons for enhancing the electrochemical performance of the WO3/SnO2 Nanocomposite Electrodes when K3Fe(CN)6 is added into a 1 M Na2SO4.

3.      Why the coulombic efficiencies of WO3/SnO2 symmetric device is very low?

4.      In WO3/SnO2 Nanocomposite electrode, which component has a major role in obtaining high-performance symmetric supercapacitors?

5.      The authors should incorporate some latest articles based on this work

https://doi.org/10.1016/j.cplett.2022.139884

https://doi.org/10.1021/acssuschemeng.1c08059

Author Response

We deeply thank to the Reviewer 1 for highly suggestive comments. We add our responses to the comments one-by-one.

------------------------------------------

Comments and Suggestions for Authors

This manuscript reports the Enhanced Performance of WO3/SnO2 Nanocomposite Electrodes with Redox-Active Electrolytes for Supercapacitors. The electrochemical activities of WO3/SnO2 Nanocomposite Electrodes was investigated in K3Fe(CN)6. K3Fe(CN)6 electrolyte. WO3/SnO2 nanocomposite electrode demonstrated excellent specific capacitance. This manuscript needs following minor revision before publication:

------------------------------------------

Comment 1

While comparing the electrochemical performance of any electrode, the electrochemical performance should be compared in similar operating conditions. The electrochemical performance of the WO3-based active electrode materials should be compared in redox active electrolytes.

Response 1

In Table 3, we compared our electrode to the other WO3-based electrodes in different conditions the authors reported. To our knowledge, the effects of redox active electrolytes on these electrodes were not yet reported. For comparison with the WO3-based electrodes, we used a WO3 electrode prepared by ourselves as a “standard”. Then, the advantage of WO3/SnO2 nanocomposite electrode and the effects of redox additives were discussed. Yes, the other WO3-based electrodes can work more in the presence of redox active electrolytes. Thus, we added short comments in the discussion and conclusion parts as below.

LINE 572: “The improved specific capacitance was also observed for single component electrodes (both of WO3 and SnO2). Therefore, this approach can be used for other electrodes using these metal oxides, although the reaction efficiency could be differed by the electrode species.”

LINE 602-: “Thus, the WO3/SnO2 nanocomposite electrode grown on carbon cloth is a promising candidates as binder free electrode for high performance of supercapacitor device, and the supercapacitor system using redox active K3Fe(CN)6 electrolyte was proposed. These strategies developed in this study can also be applied to other metal oxide nanocomposites to improve the performance of supercapacitors.”

------------------------------------------

Comment 2

Authors should explain suitable reasons for enhancing the electrochemical performance of the WO3/SnO2 Nanocomposite Electrodes when K3Fe(CN)6 is added into a 1 M Na2SO4.

Response 2

The effects of K3Fe(CN)6 could be explained by 2 factors.

  • Redox reactions of [Fe(CN)6]3-/[Fe (CN)6]4- This was confirmed by the redox peaks at 0.35 – 0.43 V and 0.18 – 0.26 V (Figure 7a), which were corresponding to the redox reactions of [Fe(CN)6]3-/[Fe (CN)6]4- couples. This could be the main reason to increase the specific capacitance.
  • Facilitating the redox reactions of WO3/SnO2. Comparing the CV curves between with and without K3Fe(CN)6 (Figure 7b), the CV curves increased their area in -0.2 – 0.2 V range. For WO3, a weak and broad reduction peak newly appeared at ~0.0 V with K3Fe(CN)6. For WO3/SnO2, the curve with K3Fe(CN)6 was almost parallel to that without K3Fe(CN)6, and a weak and broad reduction peak newly appeared at 0.0 – 0.1 V (Figure 7e). These results suggests that the enhancement in CV area was not only by the redox reaction of K3Fe(CN)6, but also due to the enhanced redox reaction of electrode materials.

These factors are explained in the manuscript with new references 57 and 58 as below.

LINE 415: “at 0.35 – 0.43 V and 0.18 – 0.26 V.”

LINE 437-442: “In details, weak and broad reduction bands newly appeared at -0.1 – 0.0 V for the WO3 electrode and 0.0 – 0.1 V for WO3/SnO2 electrode in the presence of K3Fe(CN)6 (Figure 7b, c, and e). The pseudocapacitive behavior suggested that K3Fe(CN)6 provided the additional redox reactions of the [Fe(CN)6]3-/ [Fe(CN)6]4- couple to increase the specific capacitance of these electrodes [57, 58], and also promoted the redox reactions of the WO3 component.”

------------------------------------------

Comment 3

Why the coulombic efficiencies of WO3/SnO2 symmetric device is very low?

Response 3

The GCD curves of WO3/SnO2 electrode (Figure 8) were measured at the wide potential range. At the potential higher than 1.6 V (i.e. 1.8 and 1.9 V), a redox peak of water electrolysis (*corrected) appeared in the CV curve, as explained in Line 509. Although the gas generation was not obviously observed at ~1.8 V, the water hydrolysis could consume the energy. This energy loss should decrease the coulombic efficiency of WO3/SnO2 electrode.

Using the redox active electrolyte, the reaction products ([Fe(CN)6]3- and [Fe (CN)6]4-, mutually) diffused into the whole electrolyte solution in the cell, because the electrolyte solution was not separated into anode side and cathode side. This diffusion process also decreased the availability of [Fe(CN)6]3-/[Fe(CN)6]4- ions, as the second law of thermodynamics. For the redox-flow battery, the electrolytes in cathode and anode sides should be separated to store the energy; however, in this study, we didn’t demonstrate this application.

Thus, we added the explanation as below.

LINE 529-535: “The coulombic efficiencies were low in these cases, which could be explained by a minor water hydrolysis and the diffusion of [Fe(CN)6]3-/[Fe (CN)6]4- ions. At higher potential than 1.6 V, the edge of water electrolysis started (Figure 8a), and the minor gas generation resulted in the energy loss. Using K3Fe(CN)6, the reaction products ([Fe(CN)6]3- and [Fe(CN)6]4-, mutually) diffused into the whole electrolyte solution in the cell, which decreased the availability of these ions for discharge process [62].”

------------------------------------------

Comment 4

In WO3/SnO2 Nanocomposite electrode, which component has a major role in obtaining high-performance symmetric supercapacitors?

Response 4

From Figure 5a and b, the specific capacitance of WO3 and SnO2 electrode were calculated to 240 F/g and 140 F/g at scan rate of 5 mV/s. This suggests that the WO3 was more effectively store energy than SnO2. Therefore, we consider that the WO3 was more active to store the energy. On the other hand, the WO3/SnO2 nanocomposite electrode resulted in 530 F/g at the same condition (Figure 5c). This largely improved specific capacitance of WO3/SnO2 was due to synergistic effects in morphological and electrochemical characters. Therefore, attribution of electrical performance to specific component is difficult.

------------------------------------------

Comment 5

The authors should incorporate some latest articles based on this work

Response 5

We added the references with a short comment as below.

LINE 56: New reference number [10]: https://doi.org/10.1016/j.cplett.2022.139884

LINE 77: New reference number [24]: https://doi.org/10.1021/acssuschemeng.1c08059

Reviewer 2 Report

The subject of this paper was very interesting. It was about performance of WO3/SnO2 nano-composite for supercapacitors. . As a result, the binder-free WO3/SnO2 nanocomposite with a redox-active electrolyte constituted a promising system for pseudocapacitive energy storage devices. But some points should be revised: 

- The order of the different sections should be corrected such as Materials and Method should be inserted after Introduction

- Authors should be added conclusion section after Discussion

Author Response

We deeply thank to Reviewer 2 for careful reading. We modified our manuscript according to the comments.

------------------------------------------

Comments and Suggestions for Authors

The subject of this paper was very interesting. It was about performance of WO3/SnO2 nano-composite for supercapacitors. As a result, the binder-free WO3/SnO2 nanocomposite with a redox-active electrolyte constituted a promising system for pseudocapacitive energy storage devices. But some points should be revised: 

------------------------------------------

Comment 1

- The order of the different sections should be corrected such as Materials and Method should be inserted after Introduction

Response 1

we moved the Materials and Method section after introduction.

------------------------------------------

Comment 2

- Authors should be added conclusion section after Discussion

Response 2

We added the conclusion section after Discussion as below.

Conclusions

“In this work, we developed a preparation method of WO3/SnO2 nanocomposite on carbon cloth for supercapacitor electrodes. WO3 nanospheres were first deposited on a carbon cloth substrate by a chemical bath deposition method, and then SnO2 layer was formed by an electrochemical deposition and calcination. The WO3/SnO2 nanocomposite on carbon cloth exhibited higher specific capacitance (530 F/g at 5mV/s scan rate) than those of single component electrodes (240 F/g for WO3 and 140 F/g for SnO2) in aqueous electrolyte of 1M Na2SO4. The addition of 0.02 M K3Fe(CN)6 redox active into the 1M Na2SO4 electrolyte further improved the charge storage performance of electrodes (440 F/g for WO3, 310 F/g for SnO2, and 640 F/g for WO3/SnO2 nanocomposite). These results show synergistic effecs of WO3/SnO2 nanocomposites and advantages of using redox active electrolytes for supercapacitor systems. Using a symmetric configuration of WO3/SnO2 nanocomposite electrodes, the electrolyte solution of 0.02 MK3Fe(CN)6 and 1 M Na2SO4 electrolyte demonstrated a high energy density and a power density of 64 WhKg-1 at 542 Wkg-1, respectively. The capacitive retention efficiencies for the WO3/SnO2 and WO3 electrodes with the K3Fe(CN)6 electrolyte was 94.7% after 2000 cycles at 2.5 Ag-1. Thus, the WO3/SnO2 nanocomposite electrode grown on carbon cloth is a promising candidates as binder free electrode for high performance of supercapacitor device, and the supercapacitor system using redox active K3Fe(CN)6 electrolyte was proposed. These strategies developed in this study can also be applied to other metal oxide nanocomposites to improve the performance of supercapacitors.”